# Application of minimum error entropy unscented Kalman filter in table tennis trajectory prediction

**Shenyue Luo[1], Jianfeng Niu[2], Peifeng Zheng[3], Zhihui Jing[1]\***

1 Department of Sports, University of Electronic Science and Technology of China, Chengdu, Sichuan Province, PR China, 2 Sports Coaching College, Beijing Sport University, Beijing, PR China, 3 Fujian Table Tennis, Badminton, and Tennis Management Center, Fujian Provincial Administration of Sport, Fuzhou, Fujian Province, PR China

\* 1064970914@qq.com

**Data Availability Statement:** All data files are available from the Baidu Netdisk database, and and the corresponding link is provided (https://pan.baidu.com/s/1ZmD9uyrHIDTXL7gVpb50Jw?pwd=qpaf).

## Abstract

Table tennis is important and challenging project for robotics research, and table tennis robotics receives a lot of attention from academics. Trajectory tracking and prediction of table tennis is an important technology for table tennis robots, and its estimation accuracy is also disturbed by non-Gaussian noise. In this paper, a novel Kalman filter, called minimum error entropy unscented Kalman filter (MEEUKF), is employed to estimate the motion trajectory of physical model of a table tennis. The simulation results show that the MEEUKF algorithm shows outstanding performance in tracking and predicting the trajectory of table tennis compared to some existing algorithms.

## Introduction

Table tennis is a interesting game for humans to master [1, 2], and it is also an indispensable part of human life. It is also an important and challenging project for robotics research. Accurate estimation of the trajectory of a table tennis is fundamental to the sport for robots. This paper focuses on the estimation and prediction of the trajectory of table tennis.

Accurate trajectory estimation is essential for hitting the table tennis and winning the game for robotic table tennis. The physical model of table tennis is very important for the trajectory and prediction of its trajectory. A algorithm [3] based on the forces applied approximate physical model for predicting the trajectory of a table tennis is proposed. A physical flight models that does not take into account the spin of a table tennis [4, 5]. In order to obtain more accurate tracking, the application of more accurate models in table tennis trajectory tracking studies is extremely necessary. A physical model [6, 7] that takes into account the rotation of a table tennis is applied to a table tennis robot. Some vision system [8, 9] that can predict the trajectory of spinning table tennis has been developed. The physical model of table tennis is nonlinear and the observations are often disturbed by impulse noise, which poses difficulties for the trajectory of the ball.

In recent years, information theoretic learning (ITL) is widely used in the context of state estimation problem [10–15] for deterministic models. Minimum error entropy (MEE)

**Funding:** The author(s) received no specific funding for this work.

**Competing interests:** The authors have declared that no competing interests exist.

criterion [16], in ITL, is a powerful tool in processing original data mixed with non-Gaussian noise. Moreover, the criterion is widely used in adaptive filter [17] and Kalman filter. To cope with the problem of state estimation of nonlinear system, the minimum error entropy unscented Kalman filter (MEEUKF) [12] and cubature information filter to be based on minimum error entropy (MEE-CIF) [18] are derived. These algorithms based on the MEE criterion are able to achieve outstanding performance in estimating the state of a non-linear non-Gaussian model. In this study, a physical model that takes into account the spin of table tennis is used, and we use MEEUKF to estimate this non-linear model in the presence of non-Gaussian noise. The simulation results show that the MEEUKF algorithm shows outstanding performance in tracking and predicting the trajectory of table tennis compared to some existing algorithms.

## Model

The state of the table tennis is represented by using the vector $\boldsymbol{x}_k$ with the following form

$$\boldsymbol{x}_k = [p_x, p_y, p_z, v_x, v_y, v_z, w_x, w_y, w_z]^{\mathrm{T}}, \tag{1}$$

where $p$, $v$, and $w$ denote the position, linear velocity, and angular velocity of the table tennis, respectively. $[\cdot]^{\mathrm{T}}$ represents the transpose operation of a matrix.

Assuming that only the position of the table tennis can be observed, and the observation vector can be represented as

$$\boldsymbol{z}_k = [p_x, p_y, p_z]^{\mathrm{T}}, \tag{2}$$

where $\boldsymbol{z}_k$ represents observation vector. Under ideal conditions, it is considered that the value of the angular velocity tends to be stable over a short period of time. The equation of motion for a table tennis can then be constructed as

$$\begin{cases} \begin{bmatrix} p_{x,k} \\ p_{y,k} \\ p_{z,k} \end{bmatrix} = \begin{bmatrix} p_{x,k-1} \\ p_{y,k-1} \\ p_{z,k-1} \end{bmatrix} + \begin{bmatrix} v_{x,k-1} \\ v_{y,k-1} \\ v_{z,k-1} \end{bmatrix} \Delta t + \begin{bmatrix} u_{1,k} \\ u_{2,k} \\ u_{3,k} \end{bmatrix}, \\[2em] \begin{bmatrix} v_{x,k} \\ v_{y,k} \\ v_{z,k} \end{bmatrix} = \begin{bmatrix} v_{x,k-1} \\ v_{y,k-1} \\ v_{z,k-1} \end{bmatrix} - k_1 \|V\| \begin{bmatrix} v_{x,k-1} \\ v_{y,k-1} \\ v_{z,k-1} \end{bmatrix} \Delta t \\[2em] +k_2 \begin{bmatrix} w_{x,k-1} \\ w_{y,k-1} \\ w_{z,k-1} \end{bmatrix} \begin{bmatrix} v_{x,k-1} \\ v_{y,k-1} \\ v_{z,k-1} \end{bmatrix} \Delta t - \begin{bmatrix} 0 \\ 0 \\ g \end{bmatrix} \Delta t + \begin{bmatrix} u_{4,k} \\ u_{5,k} \\ u_{6,k} \end{bmatrix}, \\[2em] \begin{bmatrix} w_{x,k} \\ w_{y,k} \\ w_{z,k} \end{bmatrix} = \begin{bmatrix} w_{x,k-1} \\ w_{y,k-1} \\ w_{z,k-1} \end{bmatrix} + \begin{bmatrix} u_{7,k} \\ u_{8,k} \\ u_{9,k} \end{bmatrix}. \end{cases} \tag{3}$$

where $[p_{x,k}, p_{y,k}, p_{z,k}, v_{x,k}, v_{y,k}, v_{z,k}, w_{x,k}, w_{y,k}, w_{z,k}]^{\mathrm{T}}$ denotes the state of the table tennis at moment $k$. $t$ denotes time interval, and $\boldsymbol{u}_k = [u_{1,k}, u_{2,k}, u_{3,k}, u_{4,k}, u_{5,k}, u_{6,k}, u_{7,k}, u_{8,k}, u_{9,k}]$ is the process noise with zero mean. $k_1 = C_D \rho A / 2m$ represents air resistance factor ($C_D$ is related to

the roughness of the ball surface, and $C_D = 0.45$ [19]; $\rho = 1.205 kg/m^3$ indicates air density; $A$ is the maximum cross-sectional area of a table tennis), $k_2$ represents the Magnus force factor [20] ($C_L = 1.23$ the lift factor; $D = 0.04m$ the diameter of the table tennis; $m = 0.0027kg$ the mass of the table tennis), $V$ indicates the magnitude of the speed of the table tennis.

The observation equation can be set as

$$\begin{bmatrix} z_{x,k} \\ z_{y,k} \\ z_{z,k} \end{bmatrix} = \begin{bmatrix} p_{x,k} \\ p_{y,k} \\ p_{z,k} \end{bmatrix} + \begin{bmatrix} r_{1,k} \\ r_{2,k} \\ r_{3,k} \end{bmatrix}. \tag{4}$$

Here, $r_k$ indicates observation noise, and observed noise is not correlated with process noise.

## UKF

According to [21], the process equation and measurement equation of non-linear system can be written as following form

$$\begin{cases} x_k = f_k(x_{k-1}) + u_{k-1}, \\ z_k = h_k(x_k) + r_{k-1}, \end{cases} \tag{5}$$

where $x_k \in \mathbb{R}^p$ stands for the system state at instant $k$, $z_k \in \mathbb{R}^q$ reoresents the measurement vector. $f_k(\cdot)$ and $h_k(\cdot)$ are the process and measurement function at instant $k$, $f_k(x_{k-1}) \in \mathbb{R}^p$, $h_k(x_k) \in \mathbb{R}^q$, $u_{k-1} \in \mathbb{R}^p$, and $r_{k-1} \in \mathbb{R}^q$ present the process and measurement Gaussian noise with covariances $Q_{k-1}$ and $R_k$ respectively. And $Q_{k-1}$ and $R_k$ are both zero means.

a) The update sigma points $X_{i,k-1|k-1}$ are evaluated by the following formulas

$$\begin{cases} X_{0,k-1|k-1} = \hat{x}_{k-1|k-1}, \\ X_{i,k-1|k-1} = \hat{x}_{k-1|k-1} \pm \left( \sqrt{(n+\lambda)P_{k-1|k-1}} \right)_i, i = 1, 2, \cdots 2p, \end{cases} \tag{6}$$

where $\hat{x}_{k-1|k-1}$ denotes the predict state at time $k-1$. $\left( \sqrt{(n+\lambda)P_{k-1|k-1}} \right)_i$ is the $i$th column vector of the square root matrix of $(n+\lambda)P_{k-1|k-1}$, $\lambda$ presents a composite scaling factor, which is defined as $\lambda = \alpha^2(p+o) - p$. Here, $\alpha$ decides the range of the sigma points, which is set as a small positive value, and parameter $o$ is always set to $3 - p$ for $p < 3$ and 0 for $p \geq 0$

b) The propagated process sigma points $X_{i,k|k-1}$ are computed by

$$X_{i,k|k-1} = f_k(X_{i,k-1|k-1}), i = 0, 1, 2, \cdots, 2p. \tag{7}$$

c) The updated state of system can be obtained by

$$x_{k|k-1} = \sum_{i=0}^{m} w_i^m X_{i,k|k-1}, \tag{8}$$

where $w_i^m$ can be obtained by

$$\begin{cases} w_0^m = \dfrac{\lambda}{p + \lambda}, \\ w_i^m = \dfrac{1}{2(n + \lambda)}, i = 1, 2, \cdots, 2p. \end{cases} \tag{9}$$

d) The error covariance $\boldsymbol{P}_{k|k-1}$ is estimated by

$$\boldsymbol{P}_{k|k-1} = \sum_{i=0}^{m} w_i^c \boldsymbol{X}_{i,k|k-1} \boldsymbol{X}_{i,k|k-1}^{\mathrm{T}} - \hat{\boldsymbol{x}}_{k|k-1} \hat{\boldsymbol{x}}_{k|k-1}^{\mathrm{T}} + \boldsymbol{Q}_k, \tag{10}$$

where $w_i^c$ can be obtained by

$$\begin{cases} w_0^c = \dfrac{\lambda}{p + \lambda} + (1 - \alpha^2 + \beta), \\ w_i^c = \dfrac{1}{2(n + \lambda)}, i = 1, 2, \cdots, 2p, \end{cases} \tag{11}$$

the parameter $\beta$ is concerned with the prior knowledge of the distribution of $\hat{\boldsymbol{x}}_{k-1|k-1}$. For Gaussian distribution, its value is set to 2.

e) The measurement sigma points $\boldsymbol{X}_{i,k|k-1}$ and the propagated measurement sigma points $\boldsymbol{Z}_{i,k|k-1}$ are estimated by

$$\begin{cases} \hat{\boldsymbol{X}}_{0,k|k-1} = \hat{\boldsymbol{x}}_{k|k-1}, \\ \hat{\boldsymbol{X}}_{i,k|k-1} = \hat{\boldsymbol{x}}_{k|k-1} \pm \left( \sqrt{(n + \lambda)P_{k|k-1}} \right)_i, i = 1, 2, \cdots, 2p, \\ \boldsymbol{Z}_{i,k|k-1} = h_k \left( \hat{\boldsymbol{X}}_{i,k|k-1} \right), i = 1, 2, \cdots, 2p. \end{cases} \tag{12}$$

f) Calculate the predicted measurement $\hat{\boldsymbol{z}}_{i,k|k-1}$

$$\hat{\boldsymbol{z}}_{i,k|k-1} = \sum_{i=1}^{m} w_i^m \boldsymbol{Z}_{i,k|k-1}. \tag{13}$$

g) The innovation covariance matrix $P_{zz,k|k-1}$ can be obtained by

$$\boldsymbol{P}_{zz,k|k-1} \approx \sum_{i=1}^{m} w_i^c \boldsymbol{Z}_{i,k|k-1} \boldsymbol{Z}_{i,k|k-1}^{\mathrm{T}} - \hat{\boldsymbol{z}}_{i,k|k-1} \hat{\boldsymbol{z}}_{i,k|k-1}^{\mathrm{T}} + \boldsymbol{R}_k. \tag{14}$$

h) Estimate the cross-covariance matrix

$$\boldsymbol{P}_{zz,k|k-1} \approx \sum_{i=1}^{m} w_i^c \boldsymbol{Z}_{i,k|k-1} \boldsymbol{Z}_{i,k|k-1}^{\mathrm{T}} - \hat{\boldsymbol{z}}_{i,k|k-1} \hat{\boldsymbol{z}}_{i,k|k-1}^{\mathrm{T}} + \boldsymbol{R}_k, \tag{15}$$

$$\boldsymbol{P}_{xz,k|k-1} \approx \sum_{i=1}^{m} w_i^c \boldsymbol{X}_{i,k|k-1} \boldsymbol{Z}_{i,k|k-1}^{\mathrm{T}} - \hat{\boldsymbol{x}}_{k|k-1} \hat{\boldsymbol{z}}_{k|k-1}^{\mathrm{T}}. \tag{16}$$

i) The Kalman gain $\boldsymbol{K}_k$ can be evaluated by

$$\boldsymbol{K}_k = \boldsymbol{P}_{xz,k|k-1}\boldsymbol{P}_{zz,k|k-1}^{-1}. \tag{17}$$

j) Get the updated state $\hat{\boldsymbol{x}}_{k|k}$ becomes

$$\hat{\boldsymbol{x}}_{k|k} = \hat{\boldsymbol{x}}_{k|k-1} + \boldsymbol{K}_k(\boldsymbol{z}_k - \hat{\boldsymbol{z}}_{k|k-1}). \tag{18}$$

## MEEUKF

We apply a statistical linearization and use measurement slope matrix $\boldsymbol{H}_k = \boldsymbol{P}_{xz,k|k-1}^{\mathrm{T}}\boldsymbol{P}_{k|k-1}^{-1}$ to solve the nonlinear estimation problem. After state prediction, we obtain the predicted state vector $\hat{\boldsymbol{x}}_{k|k-1}$ and the covariance $\boldsymbol{P}_{k|k-1}$ of predict error $\boldsymbol{\xi}_k = \boldsymbol{x}_k - \hat{\boldsymbol{x}}_{k|k-1}$. By getting $\hat{\boldsymbol{x}}_{k|k-1}$ and $\boldsymbol{\xi}_k$, we can linearise the nonlinear equation

$$\begin{aligned} \boldsymbol{z}_k &= \mathrm{h}_k(\boldsymbol{x}_k) + \boldsymbol{r}_k \\ &= \boldsymbol{H}_k\boldsymbol{\xi}_k + \mathrm{h}_k\left(\hat{\boldsymbol{x}}_{k|k-1}\right) + \boldsymbol{\varsigma}_k + \boldsymbol{r}_k, \end{aligned} \tag{19}$$

where $\boldsymbol{\varsigma}_k$ is the statistical linearization error and its covariance can be written as $\boldsymbol{F}_k = \boldsymbol{P}_{k|k-1} - \boldsymbol{P}_{xz,k|k-1}^{\mathrm{T}}\boldsymbol{P}_{k|k-1}^{-1}\boldsymbol{P}_{xz,k|k-1}$. We can get the batch mode regression form as follow:

$$\begin{bmatrix} \hat{\boldsymbol{x}}_{k|k-1} \\ \boldsymbol{z}_k + \boldsymbol{H}_k\hat{\boldsymbol{x}}_{k|k-1} - \mathrm{h}_k(\hat{\boldsymbol{x}}_{k|k-1}) \end{bmatrix} = \begin{bmatrix} \boldsymbol{I} \\ \boldsymbol{H}_k \end{bmatrix}\boldsymbol{x}_k + \begin{bmatrix} \boldsymbol{\xi}_{k-1} \\ \boldsymbol{\varsigma}_k + \boldsymbol{r}_k \end{bmatrix}, \tag{20}$$

which we set $\boldsymbol{v}_k = \begin{bmatrix} \boldsymbol{\xi}_{k-1} & \boldsymbol{\varsigma}_k + \boldsymbol{r}_k \end{bmatrix}^{\mathrm{T}}$, with

$$\begin{aligned} \mathrm{E}[\boldsymbol{v}_k\boldsymbol{v}_k^{\mathrm{T}}] &= \begin{bmatrix} \boldsymbol{P}_{k|k-1} & 0 \\ 0 & \boldsymbol{R}_k + \boldsymbol{F}_k \end{bmatrix} \\ &= \begin{bmatrix} \boldsymbol{S}_{p;k|k-1}\boldsymbol{S}_{p;k|k-1}^{\mathrm{T}} & 0 \\ 0 & \boldsymbol{S}_{r;k|k-1}\boldsymbol{S}_{r;k|k-1}^{\mathrm{T}} \end{bmatrix} \\ &= \boldsymbol{S}_k\boldsymbol{S}_k^{\mathrm{T}}. \end{aligned} \tag{21}$$

Both sides of (20) are left multiplied by $\boldsymbol{S}_k^{-1}$, and one can obtain

$$\boldsymbol{d}_k = \boldsymbol{W}_k\boldsymbol{x}_k + \boldsymbol{e}_k, \tag{22}$$

with

$$\begin{cases} \boldsymbol{d}_k = \boldsymbol{S}_k^{-1}\begin{bmatrix} \hat{\boldsymbol{x}}_{k|k-1} \\ \boldsymbol{z}_k + \boldsymbol{H}_k\hat{\boldsymbol{x}}_{k|k-1} - \mathrm{h}_k\left(\hat{\boldsymbol{x}}_{k|k-1}\right) \end{bmatrix}, \\ \boldsymbol{W}_k = \boldsymbol{S}_k^{-1}\begin{bmatrix} \boldsymbol{I} \\ \boldsymbol{H}_k \end{bmatrix}, \\ \boldsymbol{e}_k = \boldsymbol{S}_k^{-1}\boldsymbol{v}_k. \end{cases}$$

Now we introduce the MEE based cost function:

$$J(\boldsymbol{x}_k) = \frac{1}{L^2} \sum_{i=1}^{L} \sum_{i=1}^{L} G_\sigma(e_{k,i} - e_{k,j}), \tag{24}$$

where $G_\sigma(x) = \exp(-x^2/2\sigma^2)$ is Gaussian kernel function.

As part 2.2 mentioned, to minimize the error entropy, we should maximize the cost function. Calculate the derivation of $J(\boldsymbol{x}_k)$.

$$\begin{cases} \nabla J(\boldsymbol{x}_k) = \dfrac{2}{L^2\sigma} \sigma\left(W_k^{\mathrm{T}} \phi e_k - W_k^{\mathrm{T}} \varphi e_k\right) \\[2mm] [\phi]_{ij} = G_\sigma(e_k(i) - e_k(j)), i,j = 1,2,\cdots,L, \\[2mm] [\varphi]_{ij} = \begin{cases} \displaystyle\sum_{m=1}^{L} G_\sigma e_k(i) - e_k(m), i = j, \\[3mm] 0, i \neq j, i,j = 1,2,\cdots,L. \end{cases} \end{cases} \tag{25}$$

Setting $\nabla J(\boldsymbol{x}_k) = 0$, we can get the maximum state $\boldsymbol{x}_k$.

$$\begin{cases} \boldsymbol{x}_k = [\boldsymbol{W}_k^{\mathrm{T}} \boldsymbol{\chi} \boldsymbol{W}_k]^{-1} \boldsymbol{W}_k^{\mathrm{T}} \boldsymbol{\chi} \boldsymbol{d}_k, \\[2mm] \boldsymbol{\chi} = \phi - \boldsymbol{\varphi}. \end{cases} \tag{26}$$

To express $\boldsymbol{x}_k$ by $B_{pk}$ and $B_{rk}$ instead of $\boldsymbol{W}_k$, we denote:

$$\boldsymbol{\chi} = \begin{bmatrix} \boldsymbol{\chi}_{xx} & \boldsymbol{\chi}_{zx} \\ \boldsymbol{\chi}_{xz} & \boldsymbol{\chi}_{zz} \end{bmatrix}, \tag{27}$$

where $\boldsymbol{\chi}_{xx} \in \mathbb{R}^{p\times p}, \boldsymbol{\chi}_{zx} \in \mathbb{R}^{q\times p}, \boldsymbol{\chi}_{xz} \in \mathbb{R}^{p\times q}, \boldsymbol{\chi}_{zz} \in \mathbb{R}^{q\times q}$.

With a few simple derivations [11], one can obtain the state vector of system

$$\begin{cases} \hat{\boldsymbol{x}}_{k|k-1}^t = \hat{\boldsymbol{x}}_{k|k-1}^{t-1} + \bar{\boldsymbol{K}}_k\left(\boldsymbol{z}_k - \hat{\boldsymbol{z}}_{k|k-1}\right), \\[2mm] \bar{\boldsymbol{K}}_k = \left[\bar{\boldsymbol{P}}_{xx;k|k-1} + \boldsymbol{H}_k^{\mathrm{T}} \bar{\boldsymbol{P}}_{xy;k|k-1} + \left(\bar{\boldsymbol{P}}_{yx;k|k-1} + \boldsymbol{H}_k^{\mathrm{T}} \bar{\boldsymbol{P}}_{yy;k|k-1}\right)\boldsymbol{H}_k + \lambda\boldsymbol{I}\right]^{-1} \\[2mm] \times \left(\bar{\boldsymbol{P}}_{yx;k|k-1} + \boldsymbol{H}_k^{\mathrm{T}} \bar{\boldsymbol{P}}_{yy;k|k-1}\right). \end{cases} \tag{28}$$

Here,

$$\begin{cases} \bar{\boldsymbol{P}}_{xx;k|k-1} = \left(\boldsymbol{S}_{p;k|k-1}^{-1}\right)^{\mathrm{T}} \boldsymbol{\chi}_{xx} \boldsymbol{S}_{p;k|k-1}^{-1}, \\[2mm] \bar{\boldsymbol{P}}_{xy;k|k-1} = \left(\boldsymbol{S}_{r;k|k-1}^{-1}\right)^{\mathrm{T}} \boldsymbol{\chi}_{xy} \boldsymbol{S}_{p;k|k-1}^{-1}, \\[2mm] \bar{\boldsymbol{P}}_{yx;k|k-1} = \left(\boldsymbol{S}_{p;k|k-1}^{-1}\right)^{\mathrm{T}} \boldsymbol{\chi}_{yx} \boldsymbol{S}_{r;k|k-1}^{-1}, \\[2mm] \bar{\boldsymbol{P}}_{yy;k|k-1} = \left(\boldsymbol{S}_{r;k|k-1}^{-1}\right)^{\mathrm{T}} \boldsymbol{\chi}_{yy} \boldsymbol{S}_{r;k|k-1}^{-1}. \end{cases} \tag{29}$$

Here, the estimation ends as $\| \hat{\boldsymbol{x}}_{k|k-1}^t - \hat{\boldsymbol{x}}_{k|k-1}^{t-1} \| / \| \hat{\boldsymbol{x}}_{k|k-1}^t \| \leqslant \gamma$ and the updated sate $\hat{\boldsymbol{x}}_{k|k} = \hat{\boldsymbol{x}}_{k|k-1}^t$, $\gamma$ is the accuracy we want the filter can achieve. The final covariance $\boldsymbol{P}_{k|k}$ is

calculated as:

$$P_{k|k} = [I - \bar{K}_k H_k] P_{k|k-1} [I - \bar{K}_k H_k]^{\mathrm{T}} + \bar{K}_k R_k \bar{K}_k^{\mathrm{T}}. \tag{30}$$

**Algorithm 1:** MEEUKF

```
Input: The model of system,observation data zₖ
Output: Estimation of the state of system x̂ₖ|ₖ
1 Parameters setting: set proper kernel bandwidth σ; initial value
(x̂₀|₀) of the system, and an initial covariance matrix P₀|₀.
2 for i ← 1 to N do
3    Prediction of the state of the system using (6), (7), (8), and (9);
4    Prediction of the covariance of the system using (10) and (11);
5    Calculate matrix P̄ₓₓ;ₖ|ₖ₋₁, P̄ₓy;ₖ|ₖ₋₁, P̄yₓ;ₖ|ₖ₋₁, and P̄yy;ₖ|ₖ₋₁ using (29);
6    Calculate Kalman gain and update state x̂ₖ|ₖ of system utilizing (28)
7    Update covariance Pₖ|ₖ using (30)
8 end
```

## Simulation

In this part, the MEEUKF is applied to a table tennis motion model to verify the performance of the algorithm, and the effect of the parameters on the performance of the algorithm was also investigated. In the following simulations, the initial value of the table tennis state is set to [0, 0, 0, 3, 5, 5, −56, −53, 47]. All simulation results, in this paper, are averaged over 100 independent Monte Carlo runs, and in one run, 100 samples are utilised to measure the MSD that is used to measure the performance of MEEUKF algorithm.

$$MSD = \mathrm{E}[||x_k - \hat{x}_{k|k}||]. \tag{31}$$

We consider the situation where the state transition noise is Gaussian distribution and the observation noise is mixed-Gaussian and Rayleigh distribution, and these distributions of

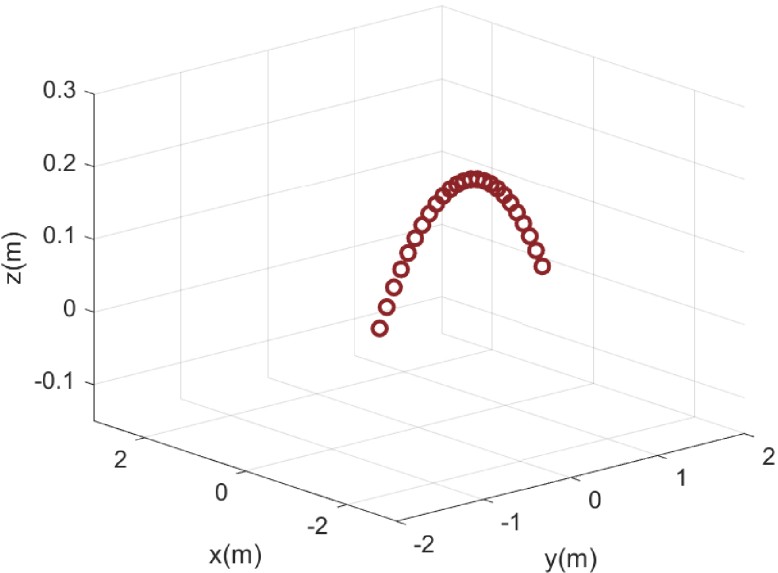

**Fig 1. The trajectory of table tennis.** The picture shows the trajectory of a table tennis.

these noises are shown below:

$$\begin{cases} \boldsymbol{u}_{k-1} \sim N(0, 0.01), \\ \boldsymbol{r}_k \sim 0.9N(0, 0.01) + 0.1N(0, 100). \end{cases} \quad (32)$$

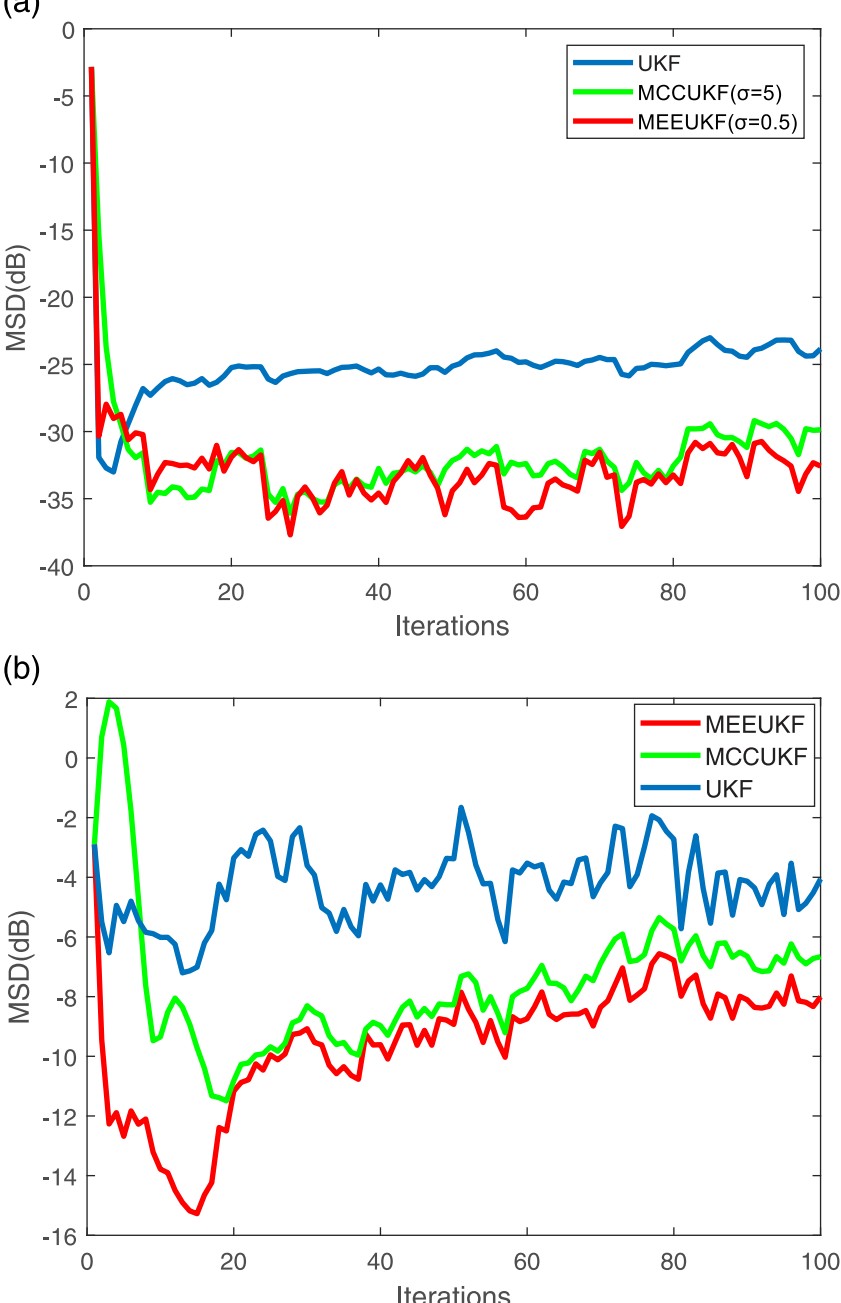

**Fig 2. Convergence curves under different noises.** The figure depicts the performance of the algorithm compared with UKF and MCCUKF in the presence of Mixed Gaussian and Rayleigh noises. (a) Mixed Gaussian noise. (b) Rayleigh noise.

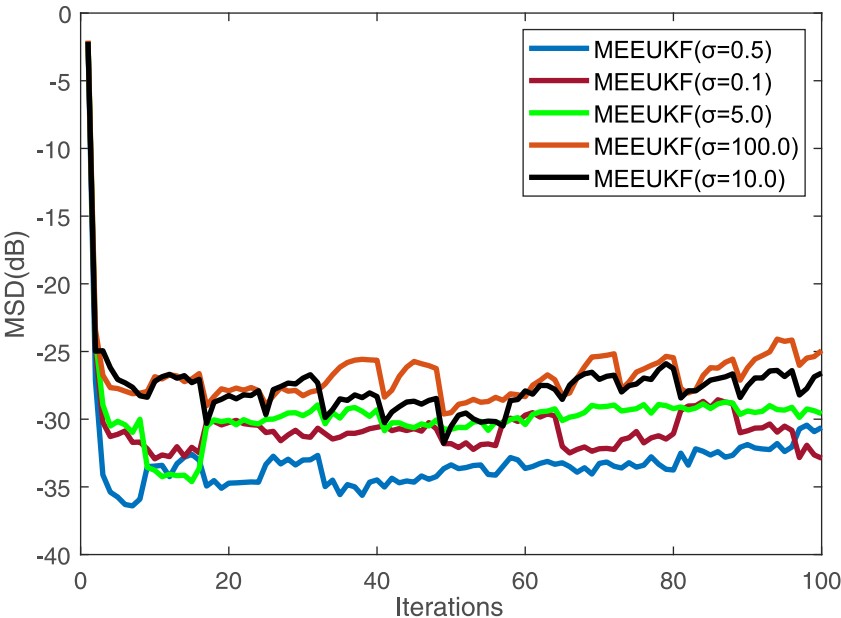

**Fig 3. Convergence curves with different kernel bandwidths γ. The picture depicts the convergence curves of the algorithm with different kernel bandwidths.**

Here, $N(\cdot, \cdot)$ denotes Gaussian distribution. Moreover, probability density function of the Rayleigh distribution is expressed as:

$$r(t) = \frac{t}{\sigma^2} \exp\left(-\frac{t^2}{2\sigma^2}\right). \tag{33}$$

The noise that obeys the Rayleigh distribution is defined as $v \sim R(\sigma)$.

First, the performance of MEEUKF applied to the table tennis motion model is compared with some existing algorithms such as UKF, MCCUKF with the presence of mixed Gaussian noise and Rayleigh noise. A table tennis motion model in the section named Model is used to test the performance of these algorithms, and the trajectory of the table tennis is shown in Fig 1. The convergence curves and parameters of these algorithms are shown in Fig 2 with the assurance that the initial convergence rate of these algorithms is almost identical. It is obvious that MEEUKF performs best when tracking table tennis in the presence of mixed Gaussian and Rayleigh noises.

Second, the effect of kernel width $\sigma$ on algorithm performance is investigated. We set different kernel width $\sigma$ = 0.1, 0.5, 5.0, 10.0, 100.0 separately to explore its impact on the performance of the algorithm, and the other parameters of the algorithm and noise are the same as in the previous simulations. The steady-state MSD and convergence curves of the algorithm are shown in Fig 3 and Table 1 respectively. One can obtain that the value of $\sigma$ is too large or

**Table 1. MSDs with different kernel bandwidths $\sigma$.**

| Bandwidths $\sigma$ | 0.1 | 0.3 | 0.5 | 1.0 | 5.0 | 10.0 | 20.0 | 50.0 | 100.0 |
|---|---|---|---|---|---|---|---|---|---|
| MSD | -31.22 | -32.82 | -34.36 | -29.14 | -20.29 | -27.35 | -26.52 | -25.9 | -25.35 |

The table depicts the relationship between the performance of the algorithm and the bandwidths $\sigma$.

too small to achieve the best performance, and the tracking performance of the MEEUKF algorithm for table tennis when $\sigma = 0.5$.

## Conclusion

For non-linear and non-Gaussian model of table tennis, the MEEUKF algorithm is used to to estimate and predict the trajectory of this model. This algorithm is able to suppress non-Gaussian noise very well. Simulations show MEEUKF performs well in predicting table tennis trajectories compared with some existing algorithms.

## Author Contributions

**Conceptualization:** Peifeng Zheng.

**Formal analysis:** Peifeng Zheng.

**Investigation:** Shenyue Luo, Jianfeng Niu.

**Methodology:** Shenyue Luo.

**Project administration:** Zhihui Jing.

**Resources:** Jianfeng Niu, Zhihui Jing.

**Software:** Shenyue Luo.

**Supervision:** Jianfeng Niu.

**Visualization:** Shenyue Luo.

**Writing – original draft:** Shenyue Luo.

**Writing – review & editing:** Zhihui Jing.

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
