## [Decision Letter · Decision Letter 0]

21 Mar 2022

PONE-D-22-05711Application of Minimum Error Entropy Unscented Kalman Filter in Table Tennis Trajectory PredictionPLOS ONE

Dear Dr. Jing,

Thank you for submitting your manuscript to PLOS ONE. After careful consideration, we feel that it has merit but does not fully meet PLOS ONE’s publication criteria as it currently stands. Therefore, we invite you to submit a revised version of the manuscript that addresses the points raised during the review process.

ACADEMIC EDITOR:  Please revise the manuscript to address all the comments from reviewers and carefully proof read the revised manuscript to improve the quality of the English writing.

We look forward to receiving your revised manuscript.

Kind regards,

Qichun Zhang, PhD

Academic Editor

PLOS ONE

Journal Requirements:

Additional Editor Comments:

The paper investigates the trajectory of table tennis using Kalman filtering. Based on my personal reading, this manuscript is well-written with solid contribution. It is a good example of generating impact as the engineering method has been adopted for modern sport training. There are some minor issues should be further revised and a proof reading is highly recommended for the resubmission. In summary, I recommend accepting this manuscript after a minor revision.

Reviewers' comments:

Reviewer's Responses to Questions

**Comments to the Author**

1. Is the manuscript technically sound, and do the data support the conclusions?

Reviewer #1: Yes

Reviewer #2: Yes

2. Has the statistical analysis been performed appropriately and rigorously? 

Reviewer #1: Yes

Reviewer #2: Yes

3. Have the authors made all data underlying the findings in their manuscript fully available?

Reviewer #1: Yes

Reviewer #2: Yes

4. Is the manuscript presented in an intelligible fashion and written in standard English?

Reviewer #1: Yes

Reviewer #2: Yes

5. Review Comments to the Author

Reviewer #1: This paper applies MEEUKF to track trajectory of table tennis, and estimates the state of the nonlinear non-Gaussian system. The simulation results show the outstanding performance of MEEUKF in tracking trajectory of table tennis. This study is very interesting and meaningful.

In general, punctuation marks need to be marked at the end of the formula, and the variable needs to be interpreted where it first appears.

For a more formal expression, “ping pong ball” should be modified to “table tennis”.

Reviewer #2: This author shows a nonlinear kinematic model of table tennis and non-Gaussian noise is considered. The MEEUKF algorithm can effectively eliminate the non-Gaussian noise in the observed data. They also show simulation results comparing MEEUKF with some existing algorithms.

In my opinion this study is very important for the development of table tennis, and this article is worthy of being published.

An algorithm flowchart should be given to better express the subject of the article, and more simulations need to be discussed.

6. PLOS authors have the option to publish the peer review history of their article (what does this mean?). If published, this will include your full peer review and any attached files.

Reviewer #1: No

Reviewer #2: No

---

## [Author Response · Author response to Decision Letter 0]

8 May 2022

Response to Reviewers

Dear Editor

 First, we would like to thank you and the reviewers for your efficient handling of our original paper. The attached manuscript is a revision of our submitted paper entitled “Application of Minimum Error Entropy Unscented Kalman Filter in Table Tennis Trajectory Prediction”. We have modified the manuscript as required to the reviewer comments. Our point-by-point replies to the comments are given below.

Reviewer #1: 

comment 1: In general, punctuation marks need to be marked at the end of the formula, and the variable needs to be interpreted where it first appears.

 Thank you for this comment. This comment can be divided into two questions. Question 1: punctuation marks need to be marked at the end of the formula; question 2: the variable needs to be interpreted where it first appears. 

 For the first question, the punctuation of these formulas (Eqs. (5), (6), (7), (8), (10), (11), (13), and (18)) is added in the corresponding parts, and the exact modifications are presented in the manuscript.

 For the second question, some variables are interpreted where it first appears, such as observation vector and time interval . Moreover, the other variables have been explained in their first appearance. 

comment 2: For a more formal expression, “ping pong ball” should be modified to “table tennis”.

 This comment is very meaningful for improving the quality of this manuscript. We have modified all the "ping pong ball" to "table tennis" in the manuscript.

Reviewer #2: 

comment 1: An algorithm flowchart should be given to better express the subject of the article, and more simulations need to be discussed.

 Thank you for this comment. This comment can be divided into two questions. Question 1: an algorithm flowchart should be given to better express the subject of the article; question 2: more simulations need to be discussed. 

 For the first question, we have added pseudo-code for the algorithm in the modified manuscript, and detailed results are shown below in manuscript.

 For the second question, an additional noise, called Rayleigh noise, is utilized to verify the performance of the algorithm. The corresponding convergence curves and necessary discussion are presented in Fig. 2(b) and manuscript, respectively.

---

## [Decision Letter · Decision Letter 1]

18 May 2022

Application of Minimum Error Entropy Unscented Kalman Filter in Table Tennis Trajectory Prediction

PONE-D-22-05711R1

Dear Dr. Jing,

We’re pleased to inform you that your manuscript has been judged scientifically suitable for publication and will be formally accepted for publication once it meets all outstanding technical requirements.

Kind regards,

Qichun Zhang, PhD

Academic Editor

PLOS ONE

Additional Editor Comments:

All the comments have been addressed well in the revised version. Therefore, I recommend accepting this paper as it is.

Reviewers' comments:

Reviewer's Responses to Questions

**Comments to the Author**

1. If the authors have adequately addressed your comments raised in a previous round of review and you feel that this manuscript is now acceptable for publication, you may indicate that here to bypass the “Comments to the Author” section, enter your conflict of interest statement in the “Confidential to Editor” section, and submit your "Accept" recommendation.

Reviewer #1: All comments have been addressed

Reviewer #2: All comments have been addressed

2. Is the manuscript technically sound, and do the data support the conclusions?

Reviewer #1: Yes

Reviewer #2: Yes

3. Has the statistical analysis been performed appropriately and rigorously? 

Reviewer #1: Yes

Reviewer #2: Yes

4. Have the authors made all data underlying the findings in their manuscript fully available?

Reviewer #1: Yes

Reviewer #2: Yes

5. Is the manuscript presented in an intelligible fashion and written in standard English?

Reviewer #1: Yes

Reviewer #2: Yes

6. Review Comments to the Author

Reviewer #1: The authors revised the manuscript adequately according to the reviewer's comments. The manuscript is now more qualified and clear. I have no further comments. I suggest accepting it for publication as it stands.

Reviewer #2: The author has fully answered my comments. So my questions have been solved.This is an outstanding with great quality.

7. PLOS authors have the option to publish the peer review history of their article (what does this mean?). If published, this will include your full peer review and any attached files.

Reviewer #1: No

Reviewer #2: No

---

## [Editor Report · Acceptance letter]

23 May 2022

PONE-D-22-05711R1 

Application of Minimum Error Entropy Unscented Kalman Filter in Table Tennis Trajectory Prediction 

Dear Dr. Jing:

I'm pleased to inform you that your manuscript has been deemed suitable for publication in PLOS ONE. Congratulations! Your manuscript is now with our production department. 

Kind regards, 

on behalf of

Dr. Qichun Zhang 

Academic Editor

PLOS ONE